# Impact of evidence-based guidelines on healthcare utilisation and costs for disc related sciatica in the Netherlands: a population-based, cross-sectional study

Juliëtte van Munster  ,[1,2] Mark W Noordenbos,[3] I J Y Halperin,[1,2] Wilbert B van den Hout,[4] Peter Paul van Benthem,[2] Ingrid Seinen,[3] Wouter A Moojen,[1,5,6] Wilco Peul[1,5]

For numbered affiliations see end of article.

**Correspondence to**
Dr Juliëtte van Munster;
j.j.c.m.van_munster@lumc.nl

## ABSTRACT

**Objective** The aim of this study was to assess the impact of high-quality evidence supporting surgical treatment of lumbar disc herniation (LDH) on healthcare practice in the Netherlands by examining changes in healthcare utilisation, including the timing of surgery, and the healthcare costs for patients with LDH.

**Design** A retrospective, cross-sectional study was performed using population-based, longitudinal data obtained from the Dutch Healthcare Authority (2007–2020) and NIVEL's primary care (2012–2020) administrative databases.

**Setting** The study was conducted within the healthcare system of the Netherlands.

**Participants** We included adults (≥18 years) who visited a Dutch hospital or a general practitioner (GP) for lumbar degenerative disc disease. Patients with LDH were identified based on registered diagnosis code, type of surgery (discectomy) and age (<56 years).

**Main outcome measures** The primary outcome measure was the difference in the annual number of LDH procedures following the publication of evidence-based guidelines in 2009 (comparing the periods 2007–2009 to 2017–2019). Secondary outcome measures focused on the timing of surgery and associated healthcare costs. To validate the outcomes, secondary outcomes also include the number of discectomies and the number of procedures in the younger age group (discectomies, laminectomies, and fusion surgery).

**Results** The number of patients suffering from LDH increased from 55 581 to 68 997 (+24%) between 2007 and 2019. A decrease was observed in the annual number of LDH procedures (−18%), in the number of discectomies (−22%) and in the number of procedures for patients aged <56 years (−18%). This resulted in lower healthcare costs by €10.5 million annually. In 2012, 31% of all patients <56 years had surgery before 12 weeks from diagnosis at the GP, whereas 20% did in 2019.

**Conclusions** Healthcare utilisation for LDH changed tremendously in the Netherlands between 2007 and 2020 and seemed to be associated with the publication and implementation of evidence-based guidelines. The observed decrease in the number of procedures has been accompanied by a corresponding reduction in healthcare costs. These findings underscore the importance of

## STRENGTHS AND LIMITATIONS OF THIS STUDY

⇒ A strength of this study is the nationwide coverage of the database.
⇒ Another strength is the fact that patients with sciatica were identified by analysing three patient identifiers (diagnosis code, age and type of surgery) to validate our findings and minimise administrative errors.
⇒ Furthermore, a strength of this study is the fact that policymakers and healthcare professionals performed this study together, supporting the independence of the study results.
⇒ A limitation of the study is that causation between guideline publication and changes in healthcare practice cannot be proven due to a lack of data availability from before 2007.

adhering to evidence-based guidelines to optimise the management of patients with LDH.

## INTRODUCTION

Lumbosacral radicular syndrome (LRS), or sciatica due to a lumbar disc herniation (LDH) is common in everyday healthcare practices with a substantial burden on healthcare and society.[1] Intolerable pain and cauda equina syndrome are definite indications for surgery, but most patients suffering from sciatica recover without the need for surgery within 3 months from onset,[2] after which period the natural course is less favourable with a longer wait-and-see strategy.[3]

In 1999, in response to a provoking practice variation publication,[4] the Health Council of the Netherlands published an evidence-based report on this topic and concluded that there was a lack of evidence on the effectiveness and timing of surgery, but also serious signs of overtreatment in the Netherlands compared with the UK.[5] In response, a randomised controlled trial was designed and performed

on this topic in the Netherlands (Sciatica trial; 2007).[6] Comparing the Netherlands with UK guidelines in the timing of discectomy, patients suffering from severe sciatica for 6–12 weeks were randomly assigned to a timing of early surgery or to prolonged conservative treatment up to 6 months with possible delayed surgery. Around the same time, the Spine Patient Outcomes Research Trial (SPORT) trial[7] was performed in the USA.[7] Both trials showed no more differences in pain scores beyond 1 year, but with faster initial pain relief and better quality-adjusted life years after early surgery compared with conservative treatment.[6 7] As a result, early surgery was a cost-effective option that does not need to be withheld for economic reasons, although it was associated with higher healthcare costs.[8]

These trials led to the development of the first multi-disciplinary, evidence-based guideline for LRS in the Netherlands in 2009[9] and several other countries.[10 11] An initial conservative treatment strategy is recommended in the first 3 months after onset of sciatica (in the absence of progressive neurological deficit and cauda equina syndrome), whereas a shared decision-making treatment strategy between patient and physician is advised after 3 months. Guideline recommendations were incorporated in 'Wise Choices' in the Dutch version of the North American initiated Choosing Wisely campaign in 2014.[12 13] In these 'Wise Choices', it is also advised to proceed to surgery within 9 months after disease onset. These recommendations were integrated in the general practitioner (GP) guideline in 2015.[14]

It was decided by the original researchers in collaboration with the Federation of Medical Specialists (FMS) and the Dutch Healthcare Authority that now is the right time to evaluate whether the evidence-based guideline recommendations changed healthcare utilisation in the Netherlands between 2007 and 2020. We focused on the number of procedures, the timing of surgical treatment for LDH-inflicted sciatica and healthcare pathways. Second, we were interested in changes in societal and healthcare costs.

## METHODS

A population-based, longitudinal, cross-sectional study was performed. The objective of this analysis was to evaluate the impact of guideline implementation on patients with sciatica attributed to LDH.

### Data sources

Two databases were analysed; first, a comprehensive nationwide, administrative healthcare database provided by the Dutch Healthcare Authority. This database contains detailed information on diagnosis by diagnosis-related groups, healthcare activities (eg, surgical procedures) and fundamental patient characteristics (eg, gender, year of birth) covering all patients treated in hospitals throughout the Netherlands.[15] Second, NIVEL's (Netherlands Institute for Health Services Research) Primary

Care Database was analysed. This database comprises routine healthcare data recorded between 2012 and 2020 and represents a 10% sample of all GP practices in the country.[16] To ensure patient confidentiality, individuals in both databases were linked using a unique anonymised identifier, thereby preventing access to sensitive personal patient data by the researchers.

### Population

This study focused on Dutch individuals aged 18 years and above who sought medical care at hospitals for sciatica due to a lumbar disc herniation between 2007 and 2020. The authors hypothesised that these patients would possess a registered diagnosis of LDH, have undergone discectomy as a treatment modality, and belong to a younger age group (<56 years) in comparison to individuals with spinal stenosis—a degenerative condition commonly observed in the elderly. This last hypothesis was based on the age distribution of patients included in the SPORT trial and Sciatica trial.[6 17] However, to mitigate potential data incompleteness, as different healthcare providers may employ varying registration practices, the initial study group incorporated multiple hospital diagnosis codes for lumbar degenerative disc disease (DDD), including the code for spinal stenosis, and corresponding surgical procedure codes (discectomy, laminectomy and instrumented spinal fusion) (online supplemental tables S1 and S2). Exclusion criteria included patients with cervical degenerative disc disease, spinal infection, traumatic or oncological fractures, spinal deformities and those who had undergone previous back surgery within the past year. From this initial study group, we selected the three subgroups for lumbar disc herniations based on the type of surgery, age and diagnosis code as stated above (for details, see online supplemental table S3). In summary, the three subgroups consisted of the following patients:

1. Lumbar disc herniation group: this group only included diagnosis codes '1360' (hernia nuclei pulposis) and '1203' (radicular syndrome), while keeping the same exclusion criteria as the initial study group. All surgical procedure codes were included in this group. These diagnosis codes are registered by orthopaedic surgeons and neurologists, respectively, after confirmation of the diagnosis. Neurologists refer patients if they might be eligible for surgery to a neurosurgeon. Orthopaedic surgeons also refer patients to neurosurgeons, but in some clinics they perform surgeries themselves or together with neurosurgeons.
2. Discectomy group: this group used the same inclusion and exclusion criteria as the initial study group, but only included surgical procedure codes for discectomy.
3. Age <56 years group: this group was based on the initial study group, but only included patients younger than 56 years. All surgical procedure codes were included in this group.

In the NIVEL database, adult patients diagnosed with lumbo-radicular syndrome (coded as L86) were included.

## Data collection

First, the annual number of hospital visits and procedures was collected. Furthermore, we collected patients' age and gender, hospital type (general hospital, (top clinical) teaching hospital, university hospital or private clinic), attending medical specialist (neurology, neurosurgery or orthopaedic surgery), date of first hospital visit, date of surgery and information on hospital waiting time for hernia-related procedures (days between order request and day of procedure). Surgical patients were assigned to the hospital where the surgery was performed. Non-surgical patients were assigned to the last hospital they visited. From the NIVEL database, the date of first visit to the GP was collected.

## Analyses

Our primary outcome included all patients with a registered diagnosis of LDH. Absolute numbers of diagnoses and procedures were calculated between 2007 and 2020. To provide a more accurate representation, surgical rates were adjusted based on the population of individuals aged 18 years and older residing in the Netherlands. Adjusted surgical rates were calculated per 10 000 inhabitants, thereby normalising the data. Additionally, we calculated in-hospital surgical rates to assess the proportion of referrals with an indication for surgical intervention. Hospital visitors for the primary outcome only included visitors with a registered LDH diagnosis code. Secondary outcomes to validate our findings included the number of discectomies and the number of procedures (discectomies, laminectomies and fusion) in the age group <56 years, for which hospital visitors included all patients with lumbar degenerative disc disease. Directly adjusted surgical rates per 10 000 inhabitants or 10 000 hospital visitors were calculated using the following formula:

$$\frac{\text{Number of surgical procedures}}{\text{Number of inhabitants} \geq 18 \text{ years / hospital visitors}} \times 10{,}000$$

Furthermore, the timing of surgical interventions was analysed by calculating the median number of days between the initial visit to the GP and the date of surgery. Lastly, we assessed the median number of days between the first visit to hospital consultation and the date of the surgical procedure.

## Cost analysis

Differences in national healthcare and societal costs were calculated by comparing the annual number of patients with LDH per 10 000 over 2007–2009 to that number over pre-COVID years 2017–2019. We updated our previous economic evaluation[8] to the population size and price level in 2022, using current surgical prices and updating other costs by the Dutch consumer price index.[15 18] Thus, the difference between a surgical and a conservative policy was estimated at €2596 in healthcare costs and €107 in societal costs per patient. With a 49% difference in surgery probability between both policies,[8] these estimates translate to slightly more than double €5298 in healthcare savings and €218 in societal savings per

prevented surgery. Societal savings are negligible because the conservative policy has higher productivity costs.

## Analytical approach

All analyses were conducted at the individual patient level. To assess differences in annual healthcare utilisation, a comparison was made between the years preceding the publication of the guidelines (2007–2009) and the most recent years (2017–2019). Due to the limited usefulness of the data from the COVID-impacted year 2020, it was included in the database but not used for direct comparisons. However, the inclusion of 2020 was deemed important to demonstrate the potential impact of the pandemic on healthcare utilisation. A larger decrease in procedures in 2020 might reflect undertreatment, which can lead to chronic disability of patients waiting too long with persistent sciatica leading to chronic disease. We chose a descriptive analysis, considering the anticipated statistical significance of even small differences given the large size of the database. All analyses were performed using R V.4.1.3.[19]

## Patient and public involvement in research

Patients were not involved in this research.

## RESULTS

### Study population

During the period from 2007 to 2020, a total of 1.9 million patients sought hospital care for DDD. The median age of these patients was 56 years and 54% were women (table 1). Among these patients, 12% underwent discectomy, laminectomy or fusion surgery (n=217 166). The most common surgical procedure was a discectomy (6%) or laminectomy (6%), while instrumented spinal fusion was less frequently used. In 53% of all cases, surgical patients were older than 56 years old and 51% were men. Teaching hospitals accounted for 52% of all procedures. Between 2012 and 2020, the NIVEL database, specifically the L86 code representing lumbo-radicular syndrome, matched with the hospital database, resulting in 52 057 patients being included in the analysis. Of these patients, 53% were women (compared with 54% in the hospital database), and the mean age was 56 years (compared with 52.4 in 2007 and 57.6 in 2020 in the hospital database). Moreover, among these patients, 16% (8267 individuals) underwent surgery, showing a slightly higher surgical rate compared with the hospital database, which reported a 12% surgical rate.

### Changes in healthcare utilisation between 2007–2009 and 2017–2019

An absolute number of procedures for sciatica patients decreased by 18% in the LDH group, 22% in the discectomy group and 18% in the age <56 years group, although an initial increase in the number of procedures was observed in all three groups between 2007 and 2010 (figure 1, online supplemental table S4). The annual

**Table 1** Baseline characteristics of the total lumbar degenerative disc disease (DDD) population between 2007 and 2020

| Characteristic | Total number of lumbar DDD patients (2007–2020, n=1 863 627 (100%*)) | Surgically-treated patients (2007–2020, n=217 166 (100%)) |
|---|---|---|
| **Age group (n, %)** | | |
| 18–45 years | 523 431 (28) | 58 648 (27) |
| 46–55 years | 391 292 (21) | 42 370 (20) |
| 56 years and older | 948 904 (51) | 116 098 (53) |
| **Gender, (n, %)** | | |
| Male | 856 732 (46) | 110 922 (51) |
| Female | 1 006 895 (54) | 106 194 (49) |
| **Diagnosis, (n, %)** | | |
| Lumbar disc herniation | 870 878 (47) | 108 036 (50) |
| Lumbar stenosis | 317 360 (17) | 65 977 (30) |
| Lumbar disc herniation and lumbar stenosis | 46 050 (2) | 31 045 (14) |
| Low back pain | 587 695 (32) | 4899 (2) |
| Unknown | 41 644 (2) | 7159 (3) |
| **Hospital type, (n, %)** | | |
| General hospital | 736 250 (40) | 63 892 (29) |
| Teaching hospital | 884 234 (47) | 113 444 (52) |
| University hospital | 94 071 (5) | 13 627 (6) |
| Private clinic | 149 072 (8) | 26 153 (12) |
| **Department, (n, %)** | | |
| Neurology department | 1 464 011 (79) | 142 786 (66) |
| Neurosurgery department | 303 626 (16) | 189 418 (87) |
| Orthopaedic surgery department | 356 674 (19) | 39 172 (18) |
| **Type of treatment†, (n, %)** | | |
| Discectomy | 107 549 (6) | 107 549 (50) |
| Laminectomy | 109 477 (6) | 109 477 (9) |
| Fusion | 20 449 (1) | 20 449 (50) |
| No surgical treatment | 1 626 152 (88) | – |
| **Referred from another hospital (n=1 777 532‡), (n, %)** | | |
| Yes | 85 164 (5) | 44 793 (26) |
| No | 1 692 368 (95) | 126 835 (74) |

*Due to rounding, the number of percentages might not add up to 100%.
†Due to the fact that patients might receive two types of surgery, the percentages might not add up to 100%.
‡Patients who visited more than one hospital on the same day were not included in this group.

number of surgical procedures for patients with LDH in private clinics increased from 719 in the years 2007–2009 to 1699 in the years 2017–2019 (figure 2, online supplemental table S4). The number of patients with LDH in teaching hospitals increased as well (+40%), whereas the numbers of surgical procedures for patients with LDH

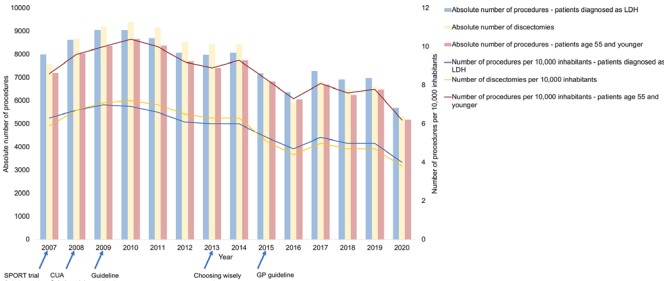

**Figure 1** Trend the absolute number of procedures and the directly standardised number of procedures; the left Y-axis and bars show the absolute number of procedures. The right Y-axis and lines show the number of procedures per 10 000 inhabitants (age>18 years). GP, general practitioner; LDH, lumbar disc herniation; SPORT, Spine Patient Outcomes Research Trial; CUA, Cost utility analysis trial.

decreased in teaching hospitals (−22%). The number of visitors and procedures for patients with LDH decreased in general hospitals (−18% and −45%, respectively) and university hospitals (−9% and −35%, respectively) in the period 2017–2019 compared with 2007–2009.

### Changes in healthcare pathways between 2007 and 2019
The number of hospital visitors diagnosed with LDH increased from 55 581 in 2007 to 68 997 in 2019, as shown in figure 3 and detailed in online supplemental table S5. Concurrently, the absolute number of patients with LDH referred by neurologists to surgical departments rose from 7155 in 2007 to 9398 in 2019, representing a 31% increase over the study period. However, the surgical rate per 10 000 LDH hospital visitors declined from 1441 per 10 000 in 2007 to 1012 per 10 000 in 2019 (figure 3, online supplemental table S4). Notably, this decrease was more pronounced among neurosurgeons, dropping from 80% in 2007 to 51% in 2019, compared with orthopaedic surgeons, whose surgical rate decreased from 11% in 2007 to 8% in 2019. In terms of referrals between hospitals for surgical and non-surgical patients with LDH, there was no substantial change observed between 2007 and 2020 across different hospital types, as depicted in figure 4.

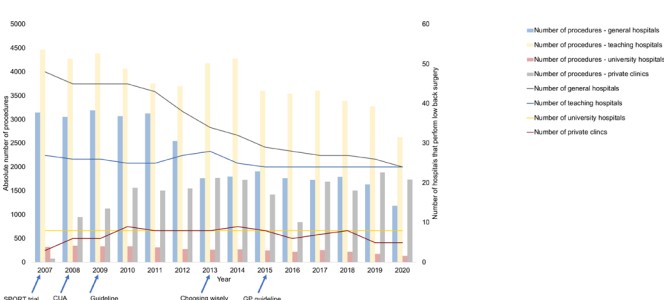

**Figure 2** Trends in absolute number of procedures per hospital type. The left Y-axis and bars show the absolute number of surgical procedures, and the right Y-axis and lines show the number of hospitals performing back surgeries. GP, general practitioner; SPORT, Spine Patient Outcomes Research Trial; CUA, Cost utility analysis trial.

van Munster J, *et al. BMJ Open* 2024;**14**:e078459. doi:10.1136/bmjopen-2023-078459

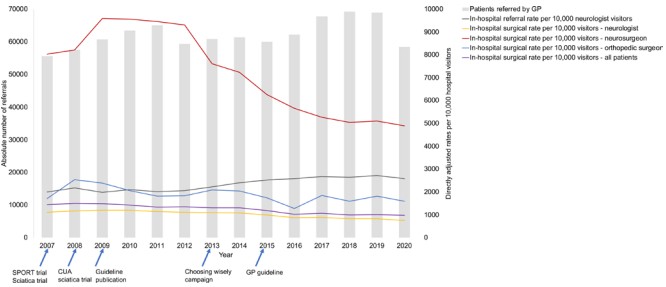

**Figure 3** GP referrals and in-hospital surgical rates. The left Y-axis and bars show the absolute number of surgical procedures, and the right Y-axis and lines show the number of procedures per 10 000 hospital visitors (in-hospital surgical rate; age 18 years and older). GP, general practitioner; SPORT, Spine Patient Outcomes Research Trial; CUA, Cost utility analysis trial.

## Timing of surgery between 2012 and 2020

The median duration in days between the initial diagnosis at the GP and the date of surgery exhibited an increase from 154 days in 2012 to 196 days in 2019, as shown in online supplemental table S6. However, the time interval between the first GP visit and the first hospital visit, also known as the time to referral, remained stable (155 days). Moreover, the days between the first hospital visit and the date of surgery increased from 71 days to 93 days (+22 days). Concurrently, the waiting time for hernia-related procedures in hospitals also increased, with an interval of 31 days in 2012 expanding to 43 days in 2019, reflecting a 12-day prolongation. Among patients below the age of 56 years, there was a decrease in the percentage of individuals who underwent surgery within 12 weeks from the

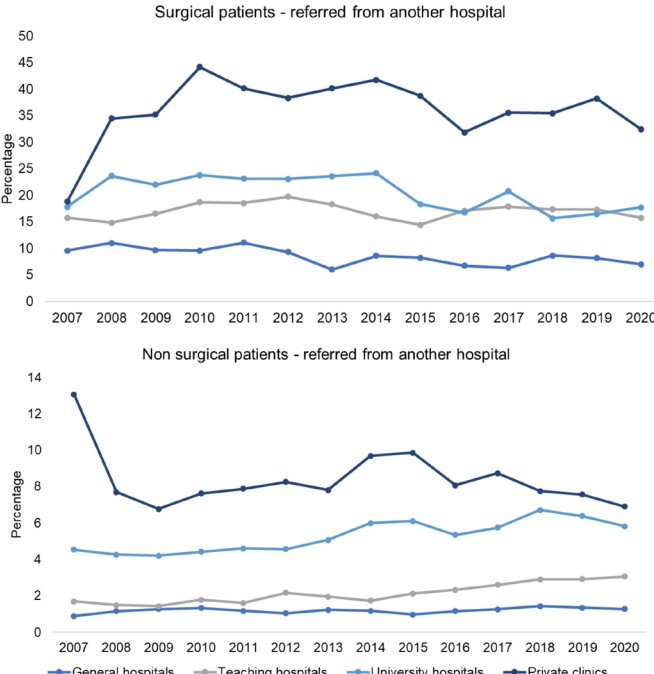

**Figure 4** Number of referrals from another hospital for surgical and non-surgical patients per hospital type. The Y-axis shows the percentage of patients that were referred from another hospital per hospital type.

time of diagnosis, declining from 31% in 2012 to 20% in 2019. Conversely, there was an increase in the proportion of patients who received surgery after 20 weeks from diagnosis, rising from 49% in 2012 to 62% in 2019. These findings highlight a shift towards longer waiting periods for surgical intervention in this patient group.

### Cost analysis

Changes in healthcare utilisation for sciatica due to LDH led to an estimated decrease of 1.5 per 10 000 inhabitants aged 18 years and older living in the Netherlands. For 2022, the associated savings were estimated at a considerable €10 million on healthcare costs and only €0.5 million on societal costs.

## DISCUSSION

Immediately following the publication of the Sciatica trial,[6] an initial increase in surgical procedures for patients with LDH was observed in the Netherlands. However, after 2010, a gradual decrease in the total number of procedures occurred, resulting in substantial annual reductions in healthcare costs and in societal costs. The largest decrease was observed between 2014 and 2016. Noteworthy changes in healthcare pathways revealed an increase in the number of referrals per surgeon, while the in-hospital surgical rate decreased. These findings suggest that both surgeons and patients are increasingly opting for an initial conservative treatment rather than early surgery.

Regarding the timing of disc surgery for sciatica, there were notable changes between 2012 and 2019. The median time interval between the initial GP visit and the date of surgery increased, resulting in 82% of all patients undergoing surgery more than 12 weeks after disease onset in 2019. It is important to note that the recommended time window for surgery, as advised in the 2009 Lumbo-Radicular Syndrome (LRS) guideline, indicating surgery at least 12 weeks after disease onset for the expectant wait-and-see period, was already met in 2012. Additionally, hospital waiting times for surgery also increased between 2012 and 2019, which does not necessarily align with the surgeon's treatment choices. Moreover, there was an increase in the percentage of patients who received surgery after more than 20 weeks. Although not outlined in the current guideline, the Dutch Choosing Wisely campaign states that surgery should be performed between 3 and 9 months after disease onset. Although some patients might experience a longer waiting time, our results indicate that most people receive surgery within 6 months after disease onset. However, recent research showed that 20 weeks after disease onset, outcomes are significantly worse in the even more prolonged conservative care group beyond the guideline time window compared with surgical treatment.[3] Therefore, careful caution in patient guidance is warranted to prevent chronic disabling sciatica, due to irreversible damage to lumbosacral nerve roots by chronic compression.[20] As this Canadian study

was published during COVID-19 times and not yet implemented in Dutch evidence-based guidelines,[3] it is not possible to do a valid investigation on the proportion of patients with a too-late indication for surgery. These study findings should be incorporated into a new set of guideline recommendations in the Netherlands.

Our findings are important because the Netherlands used to perform the second-most operations of the lumbar spine per 100 000 inhabitants compared with other high-income countries.[4] The number of procedures for LDH, however, increased in the first years after the publication of the Sciatica trial.[6] This increasing rate probably resulted from the published fact that study outcomes showed that surgery leads to faster pain relief compared with conservative treatment and was cost-effective compared with conservative care with possibly delayed surgery.[8] The decrease after 2010 might have been initiated by the publication of guideline recommendations in 2009, but also by the observation of high practice variation in surgical treatment for LDH in the Netherlands in 2010[21] leading to a task force of medical specialist federation (FMS in collaboration with ZonMW) starting the Netherlands version of the international Choosing Wisely campaign. This campaign was launched in 2013 and potentially had a huge impact since the largest decrease in the number of procedures was observed between 2014 and 2016. The active dissemination of the guidelines, and collaboration of FMS/ZonMW with insurance companies and the Patient Federation in the development of wise choices led to awareness of the favourable natural course in most sciatica patients. Not unimportantly, the growing attention to evidence-based surgery[22] within the medical specialist scientific societies, probably speeded up the implementation of the guideline in clinical practice.[23]

A notable shift in decision-making was observed among neurosurgeons and orthopaedic surgeons, who increasingly adhered to the guideline recommendations. In contrast, GPs and neurologists showed a tendency to refer even more patients with LDH than before. Furthermore, the time interval between the first GP visit and the first hospital visit remained unchanged between 2012 and 2019. This observation may indicate that patients have a preference for consulting a surgeon to discuss treatment options for LDH or that healthcare delivery for patients with LDH is increasingly organised through multidisciplinary LDH teams, as described by Michael Porter.[24] Many hospitals in the Netherlands have implemented Integrated Practice Units, so-called 'Herniated Disc-Diagnosis-Treatment Streets', where patients receive a joint appointment with both a neurologist and a surgeon, facilitating collaborative decision-making and comprehensive care.

In contrast to other hospital types, a tremendous increase in the numbers of procedures for patients with LDH was observed in private clinics. This might be attributed to a nationwide strategy aimed at redistributing and decentralising high-volume, low-complexity diagnoses such as sciatica from university and large teaching hospitals to private clinics.[25] This approach helps to concentrate resources and expertise in the larger hospitals, which primarily handle more complex cases. This resulted in a relief of pressure on the large hospitals, which was especially important during the COVID-19 pandemic: care could be maintained for these patients that required urgent or emergency neurosurgical care.[26]

A strength of this study is the nationwide coverage of the database. The fact that academic physicians performed this analysis together with policymakers from the Dutch Healthcare Authority is another strength of this study and supports the independence of the study results.

It must be mentioned, however, that two of the current supervising investigators were the Sciatica Trial's health economist (WBvdH) and principal investigator (WP), while the latter was also involved in the Guideline construction and leader of the Netherlands Choosing Wisely campaign on behalf of FMS and ZonMW. Although both researchers were initially kept out of the data analysis of the current study, some influence might be inevitable. Another limitation of this study is the fact that it is not possible to prove causation between guideline publication and changes in healthcare utilisation. Measuring the true impact of the Sciatica trial was challenging due to the unavailability of high-quality data prior to 2007. The most recent available data on the total number of lumbar disc surgeries for sciatica in the Netherlands dates back to 1996, with 11 323 patients per year.[5] Similarly, in Finland the largest reduction in discectomies occurred before 2007,[27] probably caused by the starting awareness of ongoing studies in the USA and Europe and the earlier landmark publication by Cherkin *et al* warning against overtreatment in spine surgery.[4] The last limitation of this study is the use of administrative healthcare data. The use of these data enables us to perform this investigation, but data might reflect differences in registration or coding. For example, the GP code L86 might also include lumbar stenosis. By using the three patient identifiers (ie, age, discectomy, and registered code for LDH), we tried to minimise this limitation and validate our findings.

## CONCLUSION

Although causation cannot be proven, this study highlights the substantial impact of evidence-based guidelines on clinical practice. We observed a decrease in the number of discectomy procedures, with a longer time frame between the initial visit to the general practitioner and surgery and subsequently a reduction in healthcare costs. The decrease in the number of procedures suggests that the 2008 guideline recommendations for surgical treatment for sciatica resulting from LDH were effectively implemented in clinical practice in the Netherlands, especially after the dissemination of evidence-based guidelines and the Choosing Wisely Campaign. The longer time frame observed in this study agrees with the current Dutch guideline, but recent research showed that waiting too long might lead to chronic disability. To

ensure the continued improvement of quality of care, it is essential to incorporate the findings of studies such as Bailey *et al* into evidence-based guidelines and clinical practice, thereby preventing undertreatment of patients with LDH in the future.

**Author affiliations**
[1]Neurosurgery, Leiden University Medical Center, Leiden, The Netherlands
[2]Department of Otorhinolaryngology and Head and Neck Surgery, Leiden University Medical Center, Leiden, The Netherlands
[3]Dutch Healthcare Authority, Utrecht, The Netherlands
[4]Medical Decision Making, Leiden University Medical Center, Leiden, The Netherlands
[5]Neurosurgery, Medical Center Haaglanden, The Hague, The Netherlands
[6]Neurosurgery, HAGA hospital, The Hague, The Netherlands

**Contributors** JvM designed the data analysis and data collection plan, wrote the statistical analysis plan and drafted and revised the paper. She is the guarantor. MWN analysed the data and revised the draft paper. IJYH revised the draft paper. WBvdH devised and supervised the cost analysis plan and revised the draft paper. PPvB revised the draft paper. IS monitored the data collection and analysis and revised the draft paper. WAM monitored the data analysis and data collection plan and revised the draft paper. WP initiated the collaborative project, monitored the data analysis and data collection plan and revised the draft paper.

**Funding** The Netherlands Organization for Health Research and Development (Citrienfonds), grant number: 839205002.

**Competing interests** WP: Board member of the ZonMW, former Chair of Netherlands Choosing Wisely FMS committee. PPvB: Former president of the Federation Medical Specialists (FMS). WAM: Board member of the Dutch Neurosurgical Society. WP and WBvdH were involved in the design and execution of the sciatica trial. Furthermore, WP was also involved in the guideline construction and leader of the Netherlands Choosing Wisely campaign on behalf of FMS and ZonMW.

**Patient and public involvement** Patients and/or the public were not involved in the design, or conduct, or reporting, or dissemination plans of this research.

**Patient consent for publication** Not applicable.

**Ethics approval** Our study was approved by the Medical Ethical Committee of Leiden and the Hague (metc-ldd@lumc.nl; N20.075).

**Provenance and peer review** Not commissioned; externally peer reviewed.

**Data availability statement** Data may be obtained from a third party and are not publicly available. Please see the statement of our initial submission.

**ORCID iD**
Juliëtte van Munster http://orcid.org/0000-0002-1670-1800

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
