## [Reviewer comments · BMJ Open]

ARTICLE DETAILS

TITLE (PROVISIONAL)	the impact of evidence-based guidelines on healthcare utilization and costs for disc related sciatica in the Netherlands: a population-based, cross-sectional study
AUTHORS	van Munster, Juliëtte; Noordenbos, Mark; Halperin, I.J.Y.; van den Hout, Wilbert; van Benthem, Peter Paul; Seinen, Ingrid; Moojen, Wouter; Peul, Wilco

VERSION 1 – REVIEW

REVIEWER	Furlong, Bradley Memorial University of Newfoundland, Faculty of Medicine
REVIEW RETURNED	18-Aug-2023

GENERAL COMMENTS	I think this is an interesting paper, but I felt limited in the interpretation of the results because I was uncertain about some of the inclusion and exclusion criteria around how exactly patients with lumbar disc herniations were identified. I think if the authors are more explicit in describing these criteria throughout the paper, it would strengthen the paper and facilitate better interpretation of the results. • Line 39: “The implementation of evidence-based guidelines for sciatica caused by LDH had a significant influence on healthcare practice in the Netherlands.” You rightfully pointed out in your discussion and conclusion section of the main paper that you cannot prove cause and effect from your study. However, here in your abstract’s conclusion section, it does sound like you are inferring that the guidelines (i.e., the implementation of them) had a significant effect on practice in the Netherlands. I would slightly modify the wording to make it clear that there does seem to be an association between the publication of said guidelines and subsequent practice behaviours, but I would not state this in such a manner that it sounds like the guidelines are the reason you observed such a change in practice behaviours (i.e., a decrease in LDH-associated procedures). Additionally, I would avoid using the word “significant” as this seems to infer there was some statistical test performed to show the difference was statistically significant. However, I do not see such formal statistical tests being performed in the manuscript (aside from your descriptive statistics).• This is a more general comment, but what exactly was your definition of lumbar disc herniation? I am a little confused as you state, in Table S1 for example, that you included treatment codes for pseudo radicular syndrome, radicular syndrome, spinal stenosis, DDD, etc... But there is no registered code for LDH specifically. In the databases you used, were there any additional data to indicate that these included patients had LDH? Or is it just assumed based on your hypotheses (Line 85) that they (1) had one of these treatment codes, (2) they underwent discectomy and
--

(3) they were < 56 years? Intuitively, some of these assumptions could make sense. For example, if a patient with a code of lumbar radicular syndromes receives a discectomy, then it is possible that they did have a disc herniation. But is this for certain for all the conditions listed in Table S1? For example, are there no other causes of radiculopathy aside from disc herniations? Also, I understand that < 56 years is an assumption where you hypothesize that those who are older than 56 years are more likely to have alternate conditions such as spinal stenosis, and it seems you stratified your data based on this assumption. However, I am confused as to the reasoning here because spinal stenosis is one of your inclusion criteria in Table S1. I may very well be misunderstanding your inclusion and exclusion criteria here, but regardless I do feel like it would be a good idea to more explicitly define your criteria in a few more sentences. Also, if you did not include all of the conditions listed in Table S1, please provide explanation as to why you acquired this data, as I think this is the source of much of my confusion. And again, I could be missing something here but I still think it would be a good idea to make it explicitly clear for the reader in the same section as your hypotheses. To reiterate, could you please (1) elaborate more specifically on how exactly you came to the conclusion of a patient having “a registered diagnosis of LDH” and (2) provide evidence to support these decisions? I am not necessarily disagreeing with how you’ve set up your inclusion criteria, I just think it needs to be more explicitly and clearly defined.

- Branching out from my above comment, I think it would be a good idea to discuss, possibly in your limitations section, how accurate you think your counts of LDH actually are, because as far as I understand it is possible to have spinal stenosis without LDH for example. How can you be sure that if a patient was coded as ‘spinal stenosis’, that they also had a LDH?
- Line 48: “LRS is mostly caused by a lumbar disc herniation (LDH)” Could you please add a reference for this claim, and again, if it is ‘mostly’ and not ‘always’ caused by LDHs, does this mean that your count of LDHs may not be exact as I was referring to above? If so, please discuss this somewhere in your paper.
- Line 127: “Absolute number of procedures for sciatica patients decreased between 18% and 22% over time (depending on the chosen definition)”. Could you please elaborate on these definitions? Are you referring to different definitions for sciatica? It is unclear.
- Line 135: “the surgical rate per 10,000 hospital visitors declined from 1,441 per 10,000 in 2007 to 1,012 per 10,000 in 2019.” Could you please define both (1) “surgical rate” and (2) “hospital visitors”? I am a little confused as to the surgical rate, because in your hypothesis you associate surgeries for LDH with discectomies, but then you state elsewhere you are also interested in laminectomy and fusion codes. Please define exactly what was included in your “surgery rates” in the context of this paragraph. Similarly, does hospital visitors mean any patient who visits the hospital, or solely patients who present to the hospital with an LDH-associated condition (e.g., lumbar radicular syndromes, DDD, etc.)?
- Line 102: “LDH patients per 10,000” I assume this should be “10,000”. Please revise.
- Line 118: “During the period from 2007 to 2020, a total of 1.9 million patients sought hospital care for lumbar degenerative disc disease (DDD).” My apologies if I am still misunderstanding something here, but the wording here is further confusing me as to

	what condition you are actually investigating. Are you only including patients with a DDD treatment code because it seems as though the rest of this paragraph only refers to these 1.9 million patients with DDD? What about the other patients you've included with lumbar radicular syndromes, etc. as outlined in Table S1?  • Line 171: "However, this might also be caused by the fact that data were incomplete for the years before 2011, meaning that the decrease in procedures that we observed is probably an underestimation of the true decrease in procedures." I may have missed it mentioned earlier in your paper, but why is it that pre-2011 data might be incomplete? Could you please elaborate on this a little more as this could influence the interpretation of your results, as you have already rightly suggested. It would just be nice to know why exactly the data might be incomplete so the reader is more informed. • Please add captions to your figures as they are mentioned in the text (e.g., Figure 1, Figure 2a, etc.)
--	--

REVIEWER	Cole, Ashley Sheffield Children's Hospital NHS Foundation Trust, Department of Orthopaedics & Trauma
REVIEW RETURNED	28-Aug-2023

GENERAL COMMENTS	This is a well written paper looking at national data from the Netherlands for patients with lumbar disc herniations having surgery. I think they quite rightly focus in on patients aged 18-55, the age where most patients with lumbar radicular syndrome are secondary to a disc protrusion. I am unfamiliar with the coding system used in the Netherlands but I suspect the initial GP diagnosis of lumbar disc herniation (LDH) is a clinical one without imaging to confirm and will therefore include a number of patients with referred back pain and radicular pain secondary to lateral recess stenosis. However, the number of patients having a discectomy is a good measure assuming there has been no change in coding practice? In England clinical coders were instructed to record a 'decompression' rather than a 'discectomy' if the word decompression was used in the operation note. This had the effect of significantly reducing the recorded number of lumbar discectomies with a corresponding increase in the number of discectomies. Dr Peul's RCT was published in 2007 and promoted early surgery for lumbar disc herniation. The paper refers to guidelines in 2009 suggesting that initial conservative treatment is recommended for the first 3 months which to some extent goes against the findings of the RCT 2 years before. Further guidelines with similar recommendations were published in 2014 and 2015. Table 2 shows the large reduction in the number of discectomy procedures is between 2014 and 2016 which would fit with the second set of guideline publications (probably not related to the RCT in 2007 which was not really against early surgery). Table 3 has more detailed and potentially more useful data regarding the potential causes for the reduced number of surgical procedures but only from 2012. Of note is that whilst there has been a reduction in the number of patients having surgery between 2 days and 12 weeks from first GP visit: 30.7% in 2012 to 17.2% in 2020, this has occurred very gradually. Patients having surgery in arguably the ideal time (12-20 weeks from first diagnosis) have also reduced slightly. Patients having surgery after 20 weeks from first diagnosis has increased from 49% in
---

	2012 to 68.5% in 2020. This may represent a waiting list issue or potentially an increased use of nerve root injections which is not mentioned in the paper. It certainly does not reflect an increased adherence to the guidelines. Also in Table 3, the time between the first GP visit and surgery has prolonged between 2012 (median 137 days – 19.6 weeks) and 2020 (median 230 days – 32.9 weeks). This would not be consistent with any guidelines as these times are already longer than any guideline would recommend. Page 2 line 30 >18 years should be >=18 years of >17 years according to subsequent text The conclusions state (with comments in brackets): Although causation cannot be proven, this study highlights the substantial impact of implementing high-quality randomized clinical trials and evidence-based guidelines on clinical practice (whilst I would like this to be true, I do not think the study has shown this). The findings suggest that the guideline recommendations for surgical treatment and timing of surgery for sciatica resulting from lumbar disc herniation (LDH) were effectively implemented in clinical practice in the Netherlands after 2007. This implementation led to a significant decrease in the number of discectomy procedures (as above, I don't think there was a reduction until 2014-16), with a longer timeframe between the initial visit to the general practitioner and surgery (see above, the longer timeframe is outside what would be recommended by clinical guidelines) and subsequently a reduction in healthcare costs (less surgery does reduce healthcare costs but only if the patients get better on their own rather just have surgery delayed). To ensure the continued improvement of quality of care, it is essential to incorporate the findings of studies such as Bailey et al. into evidence-based guidelines, thereby preventing undertreatment of LDH patients in the future. Summary I think this is an excellent paper and should be published but currently, I do not think the data has been interpreted correctly and the conclusions are not supported.
--	--

VERSION 1 – AUTHOR RESPONSE

Reviewer: 1

Mr. Bradley Furlong, Memorial University of Newfoundland

Comments to the Author:

I think this is an interesting paper, but I felt limited in the interpretation of the results because I was uncertain about some of the inclusion and exclusion criteria around how exactly patients with lumbar disc herniations were identified. I think if the authors are more explicit in describing these criteria throughout the paper, it would strengthen the paper and facilitate better interpretation of the results.

We thank mr. Bradley Furlong for his thorough review of our manuscript. Answers on his commentary are outlined in the marked copy and below .

- Line 39: “The implementation of evidence-based guidelines for sciatica caused by LDH had a significant influence on healthcare practice in the Netherlands.” You rightfully pointed out in your discussion and conclusion section of the main paper that you cannot prove cause and effect from

your study. However, here in your abstract's conclusion section, it does sound like you are inferring that the guidelines (i.e., the implementation of them) had a significant effect on practice in the Netherlands. I would slightly modify the wording to make it clear that there does seem to be an association between the publication of said guidelines and subsequent practice behaviours, but I would not state this in such a manner that it sounds like the guidelines are the reason you observed such a change in practice behaviours (i.e., a decrease in LDH-associated procedures). Additionally, I would avoid using the word "significant" as this seems to infer there was some statistical test performed to show the difference was statistically significant. However, I do not see such formal statistical tests being performed in the manuscript (aside from your descriptive statistics).

Answer: We agree with this comment and changed the manuscript accordingly.

Abstract: we changed the sentence "The implementation of evidence-based guidelines for sciatica caused by LDH had a significant influence on healthcare practice in the Netherlands." into the following sentence: "Healthcare utilization for LDH changed tremendously in the Netherlands between 2007 and 2020, and seemed to be associated with the publication and implementation of evidence-based guidelines."

Discussion: we changed the word significant into notable (old sentence: "Notably, a significant shift in decision-making was observed among neurosurgeons and orthopedic surgeons,;" new sentence: "A notable shift in decision-making...")

Conclusion: we changed the sentence "This implementation led to a significant decrease in the number of discectomy procedures, with a longer timeframe between the initial visit to the general practitioner and surgery and subsequently a reduction in healthcare costs." This was changed into: "We observed a decrease in the number of ... "

- This is a more general comment, but what exactly was your definition of lumbar disc herniation? I am a little confused as you state, in Table S1 for example, that you included treatment codes for pseudo radicular syndrome, radicular syndrome, spinal stenosis, DDD, etc... But there is no registered code for LDH specifically. In the databases you used, were there any additional data to indicate that these included patients had LDH? Or is it just assumed based on your hypotheses (Line 85) that they (1) had one of these treatment codes, (2) they underwent discectomy and (3) they were < 56 years? Intuitively, some of these assumptions could make sense. For example, if a patient with a code of lumbar radicular syndromes receives a discectomy, then it is possible that they did have a disc herniation. But is this for certain for all the conditions listed in Table S1? For example, are there no other causes of radiculopathy aside from disc herniations? Also, I understand that < 56 years is an assumption where you hypothesize that those who are older than 56 years are more likely to have alternate conditions such as spinal stenosis, and it seems you stratified your data based on this assumption. However, I am confused as to the reasoning here because spinal stenosis is one of your inclusion criteria in Table S1. I may very well be misunderstanding your inclusion and exclusion criteria here, but regardless I do feel like it would be a good idea to more explicitly define your criteria in a few more sentences. Also, if you did not include all of the conditions listed in Table S1, please provide explanation as to why you acquired this data, as I think this is the source of much of my confusion. And again, I could be missing something here but I still think it would be a good idea to make it explicitly clear for the reader in the same section as your hypotheses. To reiterate, could you please (1) elaborate more specifically on how exactly you came to the conclusion of a patient having "a registered diagnosis of LDH" and (2) provide evidence to support these decisions? I am not necessarily disagreeing with how you've set up your inclusion criteria, I just think it needs to be more explicitly and clearly defined.

Answer:

Table S1: the registered code for lumbar disc herniation is included in code 1360 ("Hernia Nuclei Pulposi").

The fact that we use the assumptions mentioned above was based on the fact that we know that there is variation in the registration of diagnosis codes, because these codes are mostly registered during the first consultation at the medical specialist and not always changed according to the outcomes of additional investigations. We wanted to make sure that we included all patients with a lumbar disc herniation, in order to investigate the number of discectomies (Which is the most likely procedure for a lumbar disc herniation). For example, a patient might have been (falsely) registered with spondylolisthesis, which might require decompression surgery. However, if this patient received a discectomy, we assume that the first registration was incorrect and this patient most likely had a lumbar disc herniation instead. Because this hypothesis brings along uncertainty, we validated our data by analyzing trends in the three groups that fits with the lumbar disc herniation diagnosis (age, diagnosis, surgical procedure).

To summarize:

- In the diagnosis group “LDH”, we did not include other diagnosis such as spinal stenosis. The only code included was 1360 (Hernia Nuclei Pulposi).
- In the age group and in the discectomy group, we included all diagnosis codes because we assume that age and discectomy are more likely to have a lumbar disc herniation and diagnosis codes might have been incorrectly registered.

We agree with the reviewer that this was not totally clear and changed some sections of the methods to explain this more accurately. Furthermore, it might make more sense to choose one primary outcome (change in number of procedures in LDH patients) and use the other two outcomes (discectomies, age group) to validate the findings. Mainly, the section population in the methods was extensively changed:

“Population

This study focused on Dutch individuals aged 18 years and above who sought medical care at hospitals for sciatica due to a lumbar disc herniation between 2007 and 2020. The objective of this analysis was to evaluate the impact of guideline implementation on patients with sciatica attributed to lumbar disc herniation (LDH). The authors hypothesized that these patients would possess a registered diagnosis of LDH, have undergone discectomy as a treatment modality, and belong to a younger age group (<56 years) in comparison to individuals with spinal stenosis—a degenerative condition commonly observed in the elderly. This last hypothesis was based on the age distribution of patients included in the SPORT trial and Sciatica trial⁶ 18. However, to mitigate potential data incompleteness, as different healthcare providers may employ varying registration practices, the initial study group incorporated multiple hospital diagnosis codes for lumbar degenerative disc disease (DDD), including the code for spinal stenosis, and corresponding surgical care products (discectomy, laminectomy, and instrumented spinal fusion) (Table S1 and S2). Exclusion criteria included patients with cervical degenerative disc disease, spinal infection, traumatic or oncological fractures, spinal deformities, and those who had undergone previous back surgery within the past year. From this initial study group, we selected the three subgroups for lumbar disc herniations based on type of surgery, age, and diagnosis code as stated above (For details, see table S3). Shortly, the lumbar disc herniation group was based on diagnosis code ‘1360’ (Hernia Nuclei Pulposi) using the same exclusion criteria as the initial study group and including all care products, whereas the discectomy group was based on the same inclusion and exclusion criteria for diagnosis as the initial study group, and the age group was based on the initial study group, but only patients aged younger than 56 years were included. In the NIVEL database, adult patients diagnosed with lumbo-radicular syndrome (coded as L86) were included.”

“Analyses

Our primary outcome included all patients with a registered diagnosis of LDH. Absolute numbers of diagnoses and procedures were calculated between 2007 and 2020. To provide a more accurate representation, surgical rates were adjusted based on the population of individuals aged 18 years and older residing in the Netherlands. Adjusted surgical rates were calculated per 10,000 inhabitants, thereby normalizing the data. Additionally, we calculated in-hospital surgical rates to assess the

proportion of referrals with an indication for surgical intervention. Hospital visitors for the primary outcome only included visitors with a registered LDH diagnosis code. Secondary outcomes to validate our findings included the number of discectomies and the number of procedures (discectomies, laminectomies, and fusion) in the age group < 56 years, for which hospitals visitors included all LDD patients. Directly adjusted surgical rates per 10,000 inhabitants or 10,000 hospital visitors were calculated using the following formula:”

• Branching out from my above comment, I think it would be a good idea to discuss, possibly in your limitations section, how accurate you think your counts of LDH actually are, because as far as I understand it is possible to have spinal stenosis without LDH for example. How can you be sure that if a patient was coded as ‘spinal stenosis’, that they also had a LDH?

Answer: We thank the reviewer for this suggestion. The previous section describes how we dealt with it.

• Line 48: “LRS is mostly caused by a lumbar disc herniation (LDH)” Could you please add a reference for this claim, and again, if it is ‘mostly’ and not ‘always’ caused by LDHs, does this mean that your count of LDHs may not be exact as I was referring to above? If so, please discuss this somewhere in your paper.

Answer: We removed this sentence in the new manuscript. Moreover, we have made it more clear that we counted LDHs, instead of only radicular syndrome in the diagnosis group (Table S3, method section population and the answer on the comment above).

• Line 127: “Absolute number of procedures for sciatica patients decreased between 18% and 22% over time (depending on the chosen definition)”. Could you please elaborate on these definitions? Are you referring to different definitions for sciatica? It is unclear.

Answer: we changed this line into: “Absolute number of procedures for sciatica patients decreased 18% in the LDH group, 22% in the discectomy group, and 18% in the age <56 years group...”

• Line 135: “the surgical rate per 10,000 hospital visitors declined from 1,441 per 10,000 in 2007 to 1,012 per 10,000 in 2019.” Could you please define both (1) “surgical rate” and (2) “hospital visitors”? I am a little confused as to the surgical rate, because in your hypothesis you associate surgeries for LDH with discectomies, but then you state elsewhere you are also interested in laminectomy and fusion codes. Please define exactly what was included in your “surgery rates” in the context of this paragraph. Similarly, does hospital visitors mean any patient who visits the hospital, or solely patients who present to the hospital with an LDH-associated condition (e.g., lumbar radicular syndromes, DDD, etc.)?

Answer: we more accurately defined these rates in our method section (analyses) and clarified this somehow in the result section. “Hospital visitors for the primary outcome only included visitors with a registered LDH diagnosis code. Secondary outcomes to validate our findings included the number of discectomies and the number of procedures (discectomies, laminectomies, and fusion) in the age group < 56 years, for which hospitals visitors included all LDD patients. Directly adjusted surgical rates per 10,000 inhabitants or 10,000 hospital visitors were calculated using the following formula: (Number of surgical procedures)/(Number of inhabitants ≥18 years / hospital visitors)× 10,000”

“Absolute number of procedures for sciatica patients decreased 18% in the LDH group, 22% in the discectomy group, and 18% in the age <56 years group between 18% and 22% over time (depending

on the chosen definition), although an initial increase in number of procedures was observed in all three groups between 2007 and 2010 (Figure 1a, Table 2). Annual number of surgical procedures for LDH patients in private clinics increased from 719 in the years 2007-2009 to 1699 in the years 2017-2019 (Figure 1b, Table 2). Number of LDH patients in teaching hospitals increased as well (+40%), whereas numbers of surgical procedures for LDH patients decreased in teaching hospitals (-22%). Number of visitors and procedures for LDH patients decreased in general hospitals (-18%; -45%) and university hospitals (-9%; -35%) in the period 2017-2019 compared to 2007-2009.”

• Line 102: “LDH patients per 10,0000” I assume this should be “10,000”. Please revise.

Answer: Thank you for noticing, this was revised.

• Line 118: “During the period from 2007 to 2020, a total of 1.9 million patients sought hospital care for lumbar degenerative disc disease (DDD).” My apologies if I am still misunderstanding something here, but the wording here is further confusing me as to what condition you are actually investigating. Are you only including patients with a DDD treatment code because it seems as though the rest of this paragraph only refers to these 1.9 million patients with DDD? What about the other patients you’ve included with lumbar radicular syndromes, etc. as outlined in Table S1?

Answer: since we have used three different groups (one primary, two secondary) to investigate, our total study group included all patients searching hospital care for LDDD. Therefore, we outlined this total group in the result section. We have clarified this by the changes we have made above. If this is still unclear and need to be revised, please don’t hesitate to inform us.

• Line 171: “However, this might also be caused by the fact that data were incomplete for the years before 2011, meaning that the decrease in procedures that we observed is probably an underestimation of the true decrease in procedures.” I may have missed it mentioned earlier in your paper, but why is it that pre-2011 data might be incomplete? Could you please elaborate on this a little more as this could influence the interpretation of your results, as you have already rightly suggested. It would just be nice to know why exactly the data might be incomplete so the reader is more informed.

Answer: we have removed this sentence, because this was only true for an older version of our results, in which we did not match our data with hospital’s unique patient identifier. Eventually, the Dutch Healthcare Institution (NZA) that delivered our data was able to deliver >99% of all data for the results of this article, but unfortunately we forgot to remove this sentence. Thank you for noticing. Below, we explain the data process to clarify.

Before 2011, not all fields in the dataset containing care and treatment data from Dutch hospitals were mandatory. For instance, the inclusion of a unique nationwide registration number (i.e. citizen service number) became compulsory only in 2010. Each hospital assigned a distinct patient identifier for internal use, facilitating patient identification within their specific facility. However, to monitor patients across different hospitals, a nationwide registration number is needed. To address this, a nationwide reimbursement database was utilized to enhance the care and treatment dataset with these nationwide registration numbers.

The patient matching across datasets ranged from 83% in 2007 to 96% in 2011. Among the matched patients, it was observed that approximately 95% of unique patients were accounted for based on individual hospital-specific identifiers. Moreover, these patients, on average, sought treatment at 1.04 different hospitals. In cases where there was no matching data in the reimbursement database, the hospital’s unique patient identifier was used.

Analyzing the characteristics of matched patients, it was predicted that the overestimation of patient numbers, resulting from using the hospital's unique patient identifier instead of a nationwide registration number, was less than 1% for each year between 2007 (1%) and 2011 (0.2%).

- Please add captions to your figures as they are mentioned in the text (e.g., Figure 1, Figure 2a, etc.)

Answer: thank you for noticing, we added the captions in the 'file upload step' of the submission process.

Reviewer: 2

Mr. Ashley Cole, Sheffield Children's Hospital NHS Foundation Trust

Comments to the Author:

This is a well written paper looking at national data from the Netherlands for patients with lumbar disc herniations having surgery. I think they quite rightly focus in on patients aged 18-55, the age where most patients with lumbar radicular syndrome are secondary to a disc protrusion. I am unfamiliar with the coding system used in the Netherlands but I suspect the initial GP diagnosis of lumbar disc herniation (LDH) is a clinical one without imaging to confirm and will therefore include a number of patients with referred back pain and radicular pain secondary to lateral recess stenosis.

However, the number of patients having a discectomy is a good measure assuming there has been no change in coding practice? In England clinical coders were instructed to record a 'decompression' rather than a 'discectomy' if the word decompression was used in the operation note. This had the effect of significantly reducing the recorded number of lumbar discectomies with a corresponding increase in the number of discectomies.

Dr Peul's RCT was published in 2007 and promoted early surgery for lumbar disc herniation. The paper refers to guidelines in 2009 suggesting that initial conservative treatment is recommended for the first 3 months which to some extent goes against the findings of the RCT 2 years before. Further guidelines with similar recommendations were published in 2014 and 2015. Table 2 shows the large reduction in the number of discectomy procedures is between 2014 and 2016 which would fit with the second set of guideline publications (probably not related to the RCT in 2007 which was not really against early surgery).

Table 3 has more detailed and potentially more useful data regarding the potential causes for the reduced number of surgical procedures but only from 2012. Of note is that whilst there has been a reduction in the number of patients having surgery between 2 days and 12 weeks from first GP visit: 30.7% in 2012 to 17.2% in 2020, this has occurred very gradually. Patients having surgery in arguably the ideal time (12-20 weeks from first diagnosis) have also reduced slightly. Patients having surgery after 20 weeks from first diagnosis has increased from 49% in 2012 to 68.5% in 2020. This may represent a waiting list issue or potentially an increased use of nerve root injections which is not mentioned in the paper. It certainly does not reflect an increased adherence to the guidelines.

Also in Table 3, the time between the first GP visit and surgery has prolonged between 2012 (median 137 days – 19.6 weeks) and 2020 (median 230 days – 32.9 weeks). This would not be consistent with any guidelines as these times are already longer than any guideline would recommend.

Page 2 line 30 >18 years should be >=18 years of >17 years according to subsequent text

The conclusions state (with comments in brackets):

Although causation cannot be proven, this study highlights the substantial impact of implementing high-quality randomized clinical trials and evidence-based guidelines on clinical practice (whilst I would like this to be true, I do not think the study has shown this). The findings suggest that the guideline recommendations for surgical treatment and timing of surgery for sciatica resulting from lumbar disc herniation (LDH) were effectively implemented in clinical practice in the Netherlands after 2007. This implementation led to a significant decrease in the number of discectomy procedures (as above, I don't think there was a reduction until 2014-16), with a longer timeframe between the initial visit to the general practitioner and surgery (see above, the longer timeframe is outside what would be

recommended by clinical guidelines) and subsequently a reduction in healthcare costs (less surgery does reduce healthcare costs but only if the patients get better on their own rather than just have surgery delayed). To ensure the continued improvement of quality of care, it is essential to incorporate the findings of studies such as Bailey et al. into evidence-based guidelines, thereby preventing undertreatment of LDH patients in the future.

Summary

I think this is an excellent paper and should be published but currently, I do not think the data has been interpreted correctly and the conclusions are not supported.

We thank the reviewer for his comments and answer to his comments below.

- I am unfamiliar with the coding system used in the Netherlands but I suspect the initial GP diagnosis of lumbar disc herniation (LDH) is a clinical one without imaging to confirm and will therefore include a number of patients with referred back pain and radicular pain secondary to lateral recess stenosis.

Answer: Indeed, the diagnosis is a clinical one, which might include patients with radicular pain secondary to other causes. That's why we have made the age distinction. Using administrative healthcare data, this was the best distinction we could make, but we are aware that this group might include other disease as well. We emphasized this in the strength and limitation section.

"The last limitation of this study is the use of administrative healthcare data. The use of these data enables to perform this investigation, but data might reflect differences in registration or coding. For example, the GP code L86 might also include lumbar stenosis. By using the three patient identifiers (i.e. age, discectomy, and registered code for LDH), we tried to minimize this limitation and validate our findings."

- However, the number of patients having a discectomy is a good measure assuming there has been no change in coding practice? In England clinical coders were instructed to record a 'decompression' rather than a 'discectomy' if the word decompression was used in the operation note. This had the effect of significantly reducing the recorded number of lumbar discectomies with a corresponding increase in the number of discectomies.

Answer: Indeed we can assure the reviewer that there has not been a change in the available codes in the Netherlands. Furthermore, we included not only discectomies, but also fusion surgery and laminectomy in the LDH group and the age group to validate our findings for variation in coding practices between physicians.

- Table 2 shows the large reduction in the number of discectomy procedures is between 2014 and 2016 which would fit with the second set of guideline publications (probably not related to the RCT in 2007 which was not really against early surgery).

Answer: we agree with the reviewer on this point and highlighted the fact that a large decrease happened between 2014 and 2016 in the discussion, including further interpretation.

"The decrease after 2010 might have been initiated by the publication of guideline recommendations in 2009, but also by the observation of high practice variation in surgical treatment for LDH in the Netherlands in 2010²² leading to a task force of the medical specialist federation (FMS in collaboration with ZonMW) starting the Netherlands version of the international Choosing Wisely campaign. This campaign was launched in 2013 and potentially had a huge impact, since the largest

decrease in number of procedures was observed between 2014 and 2016.”

- Patients having surgery after 20 weeks from first diagnosis has increased from 49% in 2012 to 68.5% in 2020. This may represent a waiting list issue or potentially an increased use of nerve root injections which is not mentioned in the paper. It certainly does not reflect an increased adherence to the guidelines.
Also in Table 3, the time between the first GP visit and surgery has prolonged between 2012 (median 137 days – 19.6 weeks) and 2020 (median 230 days – 32.9 weeks). This would not be consistent with any guidelines as these times are already longer than any guideline would recommend.

We agree with the reviewer that the prolonged time between first GP visit and surgery might be due to other issues as we already described in the discussion. A time frame is described in the Dutch Wise choices (between 3 and 9 months), however, in the Dutch guideline, no maximum time-frame has yet been described. Hopefully, like we stated in our conclusion, the revised Dutch guideline will also incorporate the study from Bailey et al., leading to actions to treat HNP patients within the right timeframe. We agree with the reviewer, that a longer time-frame doesn't necessarily indicate good clinical care and therefore changed our statement in the conclusion. We included the fact that we did not analyze nerve root injections, as this might indeed lead to longer waiting times. Furthermore, we clarified these findings in the discussion.

- Page 2 line 30 >18 years should be ≥18 years of >17 years according to subsequent text

Answer: we changed this in the text, thank you for noticing.

- The conclusions state (with comments in brackets):
Although causation cannot be proven, this study highlights the substantial impact of implementing high-quality randomized clinical trials and evidence-based guidelines on clinical practice (whilst I would like this to be true, I do not think the study has shown this). The findings suggest that the guideline recommendations for surgical treatment and timing of surgery for sciatica resulting from lumbar disc herniation (LDH) were effectively implemented in clinical practice in the Netherlands after 2007. This implementation led to a significant decrease in the number of discectomy procedures (as above, I don't think there was a reduction until 2014-16), with a longer timeframe between the initial visit to the general practitioner and surgery (see above, the longer timeframe is outside what would be recommended by clinical guidelines) and subsequently a reduction in healthcare costs (less surgery does reduce healthcare costs but only if the patients get better on their own rather just have surgery delayed). To ensure the continued improvement of quality of care, it is essential to incorporate the findings of studies such as Bailey et al. into evidence-based guidelines, thereby preventing undertreatment of LDH patients in the future.

Answer: we revised the conclusion section of our study, to make it more accurate and in line with our study results.

“Although causation cannot be proven, this study highlights the substantial impact of evidence-based guidelines on clinical practice. We observed a decrease in the number of discectomy procedures, with a longer timeframe between the initial visit to the general practitioner and surgery and subsequently a reduction in healthcare costs. The decrease in number of procedures suggest that the 2008 guideline recommendations for surgical treatment for sciatica resulting from lumbar disc herniation (LDH) were effectively implemented in clinical practice in the Netherlands, especially after the dissemination of evidence-based guidelines and the Choosing Wisely Campaign. The observed longer-time frame observed in this study, agrees with the current Dutch guideline, but recent research showed that

waiting too long might lead to chronic disability. To ensure the continued improvement of quality of care, it is essential to incorporate the findings of studies such as Bailey et al. into evidence-based guidelines and clinical practice, thereby preventing undertreatment of LDH patients in the future.”

Reviewer: 1

Competing interests of Reviewer: None

Reviewer: 2

Competing interests of Reviewer: I was involved in the development of the UK National Low Back and Radicular Pain Pathway as recommended by NICE NG59). I am the current

Editor. https://www.ukssb.com/files/ugd/dd7c8a_caf17c305a5f4321a6fca249dea75ebe.pdf

VERSION 2 – REVIEW

REVIEWER	Furlong, Bradley Memorial University of Newfoundland, Faculty of Medicine
REVIEW RETURNED	27-Nov-2023

GENERAL COMMENTS	Thank you for your detailed responses. You have satisfactorily addressed my comments and it is much easier to interpret the data with your elaboration on the three subgroups. I think the changes have considerably strengthened your paper and I have no further comments.
--

REVIEWER	Cole, Ashley Sheffield Children’s Hospital NHS Foundation Trust, Department of Orthopaedics & Trauma
REVIEW RETURNED	03-Dec-2023

GENERAL COMMENTS	I would thank the authors for the changes made which have improved the paper significantly. I now understand the groups of patients and the issues of coding:  1. LDH Group: The LDH group have a diagnosis of HNP ‘1360’. However, these patients may have undergone discectomy, laminectomy or fusion. This group includes patients with a misdiagnosis of LDH and will not include patients who had discectomy but were given a different (incorrect) diagnosis. 2. Discectomy Group: The discectomy group largely ignores the diagnosis but just includes those having a lumbar discectomy). 3. Age <56 group: The Age <56 includes all lumbar diagnoses in addition to all lumbar procedures (discectomy, laminectomy and fusion) so would included those having surgery for spondylolisthesis, stenosis, fusion for back pain etc. The primary analysis was done on the LDH group so I think they do need to state who makes this diagnosis and when in the treatment pathway it is made. Perhaps a list explaining the 3 groups as above. I suppose I still have a preference for the discectomy group as surgical procedure is usually more accurately recorded than diagnosis. I cannot see the value in the ‘Age <56’ group as this includes too many diagnoses and procedures. The term ‘care products’ is an unusual one – maybe ‘surgical procedures’ would be better?
--

VERSION 2 – AUTHOR RESPONSE

Reviewer: 1

Mr. Bradley Furlong, Memorial University of Newfoundland

Comments to the Author:

Thank you for your detailed responses. You have satisfactorily addressed my comments and it is much easier to interpret the data with your elaboration on the three subgroups. I think the changes have considerably strengthened your paper and I have no further comments.

We thank Mr. Bradley Furlong for his thorough review of our manuscript. We are excited to hear that our changes are satisfactory.

Reviewer: 2

Mr. Ashley Cole, Sheffield Children's Hospital NHS Foundation Trust

Comments to the Author:

I would thank the authors for the changes made which have improved the paper significantly.

I now understand the groups of patients and the issues of coding:

1. LDH Group: The LDH group have a diagnosis of HNP '1360'. However, these patients may have undergone discectomy, laminectomy or fusion. This group includes patients with a misdiagnosis of LDH and will not include patients who had discectomy but were given a different (incorrect) diagnosis.
2. Discectomy Group: The discectomy group largely ignores the diagnosis but just includes those having a lumbar discectomy).
3. Age <56 group: The Age <56 includes all lumbar diagnoses in addition to all lumbar procedures (discectomy, laminectomy and fusion) so would include those having surgery for spondylolisthesis, stenosis, fusion for back pain etc.

The primary analysis was done on the LDH group so I think they do need to state who makes this diagnosis and when in the treatment pathway it is made. Perhaps a list explaining the 3 groups as above.

I suppose I still have a preference for the discectomy group as surgical procedure is usually more accurately recorded than diagnosis. I cannot see the value in the 'Age <56' group as this includes too many diagnoses and procedures.

The term 'care products' is an unusual one – maybe 'surgical procedures' would be better?

We thank Mr. Ashley Cole for his in-depth review of our manuscript. Answers on his remarks are outlined in the marked copy and below.

LDH Group: The LDH group have a diagnosis of HNP '1360'. However, these patients may have undergone discectomy, laminectomy or fusion. This group includes patients with a misdiagnosis of LDH and will not include patients who had discectomy but were given a different (incorrect) diagnosis.

We added a diagnosis code in Table S3 which we erroneously did not include in the previous version of the table in the included diagnosis codes for the lumbar disc herniation diagnosis group ("0330-12-00-1203 – Radicular syndrome").

The primary analysis was done on the LDH group so I think they do need to state who makes this diagnosis and when in the treatment pathway it is made. Perhaps a list explaining the 3 groups as above.

Answer: We agree with the reviewer that it could be more clear how the treatment pathway is orchestrated and what patients our different study groups consist of. We thank the reviewer for his suggestion.

We revised the description of the different groups in the Population section of the Methods:

“In summary, the three subgroups consisted of the following patients:

1) Lumbar disc herniation group: this group only included diagnosis codes ‘1360’ (Hernia Nuclei Pulposi) and ‘1203’ (Radicular syndrome), while keeping the same exclusion criteria as the initial study group. All surgical procedure codes were included in this group.

These diagnosis codes are registered by orthopedic surgeons and neurologist respectively after confirmation of the diagnosis. Neurologists refer patients if they might be eligible for surgery to a neurosurgeon. Orthopedic surgeons also refer patients to neurosurgeons, but in some clinics they perform surgeries themselves or together with neurosurgeons.

2) Discectomy group: this group used the same inclusion and exclusion criteria as the initial study group, but only included surgical procedure codes for discectomy.

3) Age <56 years group: this group was based on the initial study group, but only included patients younger than 56 years. All surgical procedure codes were included in this group.”

I suppose I still have a preference for the discectomy group as surgical procedure is usually more accurately recorded than diagnosis. I cannot see the value in the ‘Age <56’ group as this includes too many diagnoses and procedures.

Answer: Although the reviewer has made a valid point that surgical procedure codes are more accurately recorded than diagnosis, we still are in agreement that the addition of the ‘Age <56’ group is valuable for the interpretation of our study results. We refer to line 92-97:

“The authors hypothesized that these patients would possess a registered diagnosis of LDH, have undergone discectomy as a treatment modality, and belong to a younger age group (<56 years) in comparison to individuals with spinal stenosis—a degenerative condition commonly observed in the elderly. This last hypothesis was based on the age distribution of patients included in the SPORT trial and Sciatica trial^{6 18}. However, to mitigate potential data incompleteness, as different healthcare providers may employ varying registration practices, the initial study group incorporated multiple hospital diagnosis codes for lumbar degenerative disc disease (DDD), including the code for spinal stenosis, and corresponding surgical procedure codes (discectomy, laminectomy, and instrumented spinal fusion) (Table S1 and S2)”

We were forced to either include too few treatment codes which could lead to underrepresentation of LDH patients in the data and thus leading to invalidity, or include more treatment codes resulting in a more valid representation of these patients in the data. As the reviewer comments, this could result in other diagnoses and procedures being included, decreasing the accuracy of the study results. We chose to include all diagnosis and treatment codes, as we agreed that most patients <56 years would probably be diagnosed with LDH and not other degenerative back diseases. Therefore, we hypothesized that the effect on the accuracy of the data is not significant, while we increased the validity of the data. It is true that other types of surgery may be included, however, we feel that the number of fusion surgeries are negligible for LDH patients <56 years and laminectomy registration codes may also include a performed discectomy as explained in the citation above.

In conclusion, by adding a <56 years group, we have a patient group consisting of mostly LDH patients for which discectomy and laminectomy are the most performed surgeries, which can be coded in multiple way. If we do not add this group to our study, we would not be able to show that especially for LDH patients <56 years of age, the surgery rate per 10.000 inhabitants as well as per 10.000 hospital visitors has diminished considerably. We also would not be able to show that the time till surgery has increased in this patient group over the years, leading to an increase in patients having

surgery >20 weeks after diagnosis at a general practitioner's office, potentially leading to chronic disabling sciatica in this patient group.

The term 'care products' is an unusual one – maybe 'surgical procedures' would be better?

We agree that this term might be confusing for readers who are not familiar with the Dutch healthcare coding system. Therefore, we changed 'care products' into 'surgical procedure codes' where applicable in the main text and the supplemental tables.

Reviewer: 1

Competing interests of Reviewer: None

Reviewer: 2

Nil new

VERSION 3 – REVIEW

REVIEWER	Cole, Ashley Sheffield Children's Hospital NHS Foundation Trust, Department of Orthopaedics & Trauma
REVIEW RETURNED	21-Jan-2024
GENERAL COMMENTS	The authors have dealt with my minor issues very well and I think the manuscript is now very clear and readers can make their own informed interpretation. Happy to proceed with publication